# Phenotypic Characterization of Circulating Tumor Cells Isolated from Non-Small and Small Cell Lung Cancer Patients

**DOI:** 10.3390/cancers15010171

**Published:** 2022-12-28

**Authors:** Argyro Roumeliotou, Evangelia Pantazaka, Anastasia Xagara, Foteinos-Ioannis Dimitrakopoulos, Angelos Koutras, Athina Christopoulou, Theodoros Kourelis, Nada H. Aljarba, Saad Alkahtani, Filippos Koinis, Athanasios Kotsakis, Galatea Kallergi

**Affiliations:** 1Laboratory of Biochemistry/Metastatic Signaling, Section of Genetics, Cell Biology and Development, Department of Biology, University of Patras, 26504 Patras, Greece; 2Laboratory of Oncology, Faculty of Medicine, School of Health Sciences, University of Thessaly, 41500 Larissa, Greece; 3Division of Oncology, Department of Medicine, University Hospital, Medical School, University of Patras, 26504 Patras, Greece; 4Oncology Unit, ST Andrews General Hospital of Patras, 26332 Patras, Greece; 5Department of Medical Oncology, “Olympion” General Hospital, 26443 Patras, Greece; 6Department of Biology, College of Science, Princess Nourah bint Abdulrahman University, P.O. Box 84428, Riyadh 11671, Saudi Arabia; 7Department of Zoology, College of Science, King Saud University, P.O. Box 2455, Riyadh 11451, Saudi Arabia; 8Department of Medical Oncology, University General Hospital of Larissa, 41221 Larisa, Greece

**Keywords:** circulating tumor cells, non-small-cell lung cancer, small-cell lung cancer, lung cancer, JUNB, CXCR4

## Abstract

**Simple Summary:**

Studies have shown that JUNB and CXCR4 contribute to cell proliferation, migration and invasion, hence claiming an important role in tumor progression and metastasis, in various cancer types, including lung cancer. We have previously reported that JUNB and CXCR4 are overexpressed in circulating and disseminated tumor cells from breast cancer patients and are correlated with worse survival. In the present study, we investigated the expression of JUNB and CXCR4 in circulating tumor cells (CTCs) of non-small-cell lung cancer and small-cell lung cancer patients and determined their clinical significance. Our results showed that both JUNB and CXCR4 were overexpressed in CTCs from lung cancer patients. Furthermore, both proteins were correlated with poor survival in non-small-cell lung cancer patients, while only CXCR4 was associated with worse survival in small-cell lung cancer patients, suggesting that JUNB and CXCR4 are potentially important prognostic biomarkers for lung cancer.

**Abstract:**

In the present study, we evaluated the expression of JUNB and CXCR4 in circulating tumor cells (CTCs) of lung cancer patients and investigated whether these proteins have prognostic clinical relevance. Peripheral blood from 30 patients with non-small-cell lung cancer (NSCLC) was filtered using ISET membranes, and cytospins from 37 patients with small-cell lung cancer (SCLC) were analyzed using confocal and VyCAP microscopy. Both JUNB and CXCR4 were expressed in the vast majority of lung cancer patients. Interestingly, the phenotypic patterns differed between NSCLC and SCLC patients; the (CK+/JUNB+/CXCR4+) phenotype was present in 50% of NSCLC vs. 71% of SCLC patients. Similarly, the (CK+/JUNB+/CXCR4–) was present in 44% vs. 71%, the (CK+/JUNB–/CXCR4+) in 6% vs. 71%, and the (CK+/JUNB–/CXCR4–) phenotype in 38% vs. 84%. In NSCLC, the presence of ≥1 CTCs with the (CK+/JUNB+/CXCR4+) phenotype was associated with worse progression-free survival (PFS) (*p =* 0.007, HR = 5.21) while ≥2 with poorer overall survival (OS) (*p* < 0.001, HR = 2.16). In extensive stage SCLC patients, the presence of ≥4 CXCR4-positive CTCs was associated with shorter OS (*p =* 0.041, HR = 5.01). Consequently, JUNB and CXCR4 were expressed in CTCs from lung cancer patients, and associated with patients’ survival, underlying their key role in tumor progression.

## 1. Introduction

Lung cancer has the highest mortality rate and frequency among cancers, affecting millions of patients, on a worldwide scale [1,2,3]. The two major types of lung cancer are non-small-cell lung cancer (NSCLC), comprising approximately 85% of total lung cancer cases, and small-cell lung cancer (SCLC), accounting for the rest of about 15% of cases [1,3,4,5]. High metastatic potential, early relapse and eventually death, characterize lung cancer [1,3,4,6]. In addition, SCLC is the most aggressive type and is usually characterized by higher numbers of circulating tumor cells (CTCs) in patients’ blood [5,7].

Patients with lung cancer are usually diagnosed at an advanced stage [6], and samples, obtained through solid biopsy, are difficult to collect [8]. Liquid biopsy offers higher accessibility regarding sample collection, and CTCs represent a more comprehensive view of cancer’s molecular and cellular heterogeneity, encompassing information from both the primary tumor and/or the metastatic lesions [2,7,8]. In the last two decades, CTCs’ enumeration and phenotypic characterization have demonstrated encouraging results regarding the prognosis of cancer patients and have provided useful biomarkers for improved diagnosis and potential future targeted therapies [1,7,8,9]. On the other hand, it has been reported that CTCs, especially in lung cancer, may express low levels of epithelial markers [1,7,9,10,11] raising difficulties upon their detection. One possible mechanism responsible for the loss of epithelial markers in CTCs is epithelial-to-mesenchymal transition (EMT) [11,12,13,14,15]. Moreover, there are studies indicating that populations of CTCs may express stem-like markers [4,16,17]. These observations further strengthen the need for new biomarkers in CTCs to improve their role in clinical practice.

Earlier studies of our group in CTCs [18] and disseminated tumor cells (DTCs) [19] from breast cancer patients identified a new pair of biomarkers, namely the C-X-C motif chemokine receptor 4 (CXCR4) and the transcription factor JUNB. CXCR4 is involved in homeostasis through leukocyte trafficking and regulation of hematopoiesis, but also participates in cancer progression and metastasis in several neoplasms, including lung cancer [20,21]. The transcription factor JUNB is essential in cell proliferation and differentiation during development but is also involved in invasion and metastasis [22,23,24], and is implicated in pathways related to EMT, in several neoplasms including lung cancer [25,26,27].

Our studies have shown that JUNB and CXCR4 are significantly overexpressed in CTCs isolated from metastatic breast cancer patients compared to normal donors’ and patients’ peripheral blood mononuclear cells (PBMCs). The presence of JUNB was associated with significantly lower median PFS and OS. JUNB has also emerged as an independent prognostic factor for OS [18]. Previous reports have shown that CXCR4 was expressed in NSCLC and associated with decreased patients’ survival [20,28], while others have demonstrated that CXCR4 expressed in NSCLC was related to EMT and stem-like characteristics [16,29]. Furthermore, CXCR4 was overexpressed in CTCs from SCLC patients at extensive-stage disease, before the initiation of chemotherapy, where its presence was prognostic for shorter PFS [30].

CTCs entering the lymph and blood can be transferred in the bone marrow, where they can survive as DTCs and subsequently form metastasis. We have previously also shown that DTCs isolated from the bone marrow of early, non-metastatic, breast cancer patients also express high levels of JUNB and CXCR4, with their expression being similar or higher to other breast cancer cell lines [19]. The phenotype (CK+/JUNB+/CXCR4+) was the most frequent among DTCs and was associated with lower OS [19].

In the present study, we evaluated, for the first time, the expression of JUNB and CXCR4 in CTCs from patients with NSCLC and SCLC and examined whether their expression was associated with clinical outcomes.

## 2. Materials and Methods

### 2.1. Cancer Cell Culture

The H1299 cell line was obtained from the American Type Culture Collection (Manassas, VA, USA) and was used for the evaluation of JUNB and CXCR4 expression. H1299 cells were cultured in Dulbecco’s Modified Eagle Medium with Glutamax (Thermo Fisher Scientific, Waltham, MA, USA) supplemented with 10% fetal bovine serum (FBS; PAN-Biotech, Germany), and 50 U/mL penicillin/50 μg/mL streptomycin (Thermo Fisher Scientific, Waltham, MA, USA). Cells were maintained at 37 °C in a humidified atmosphere of 5% CO_2_ in air. Sub-cultivation was performed with 0.25% trypsin-EDTA (Thermo Fisher Scientific, Waltham, MA, USA).

### 2.2. Patients’ Characteristics and Patients’ Blood Specimen Collection

A total of 30 chemotherapy-naïve patients with locally unresectable or metastatic NSCLC were enrolled in this study, before the administration of frontline chemotherapy (at baseline). Patients’ median age was 66 years old (range: 44–82). Most patients were at stage IV (80%). The majority of tumors were characterized histologically as adenocarcinomas (60%) and the rest as squamous cell carcinomas (40%) (Table 1).

Furthermore, a total of 37 chemotherapy-naïve SCLC patients were enrolled in the study. Patients’ median age was 68 years old (range: 44–79) and 54% of them had extensive disease. Most patients (78%) were smokers or ex-smokers. (Table 2).

Peripheral blood (10 mL), from all patients as well as from 4 healthy donors, was collected in EDTA K2 tubes. All blood samples were collected at the middle of vein puncture, as long as the first 5 mL were discarded, in order to avoid contamination with epithelial cells from the skin. The protocol was approved by the Ethics and Scientific Committees of all the involved institutions: University General Hospital of Heraklion, 71500 Heraklion, Crete, Greece (Ethical Allowance: 8756/23-6-2014, approval date: 6 August 2014); Metropolitan General Hospital, 15562 Athens (35/00-03/16); University General Hospital of Larissa, 41334 Larissa, Greece (32710/3-8-20); University General Hospital of Patras, 26504 Patras, Greece and Olympion Hospital-General Clinic of Patras, 26443 Patras, Greece (172/18-09-2020); ST Andrews General Hospital of Patras, 26332 Patras, Greece (521/13-10-2020). Patients and healthy donors gave their informed written consent for having their blood collected and for their clinical follow-up data to be used for research purposes.

### 2.3. ISET Isolation

Isolation of CTCs and all experimental procedures were performed immediately after collection or at most within 24 h of blood collection. Isolation of CTCs from NSCLC is challenging because of the extensive loss of epithelial markers on their surface and their fewer numbers in the blood [1,7,9]. Due to these difficulties, isolation of CTCs from NSCLC blood samples was performed using the ISET (isolation based on the size of tumor cells) technology. The method, which is very well established in our lab according to our previous publications [11,31] is label-independent and can successfully capture CTCs which present heterogeneous phenotypic profiles. Isolation/filtration relies on the larger size of CTCs compared to the majority of leukocytes, increasing the recovery rate of CTCs [31]. CTCs were isolated according to the manufacturer’s instructions. For this purpose, 10 mL of peripheral blood in EDTA tubes was diluted in 1:10 ISET buffer (Rarecells, Paris, France) for 10 min at room temperature (RT). This application-specific buffer performs erythrolysis and contains formaldehyde to preserve the integrity of CTCs. The diluted samples were filtered through the ISET membrane using depression adjusted at 10 kPa, whereby fixed cells that cannot pass through the pores (i.e., CTCs) are retained on the membrane, forming 10 spots. The method’s detection threshold based on the manufacturing company is 1 CTC per 10 mL of blood. We have previously evaluated CTCs’ recovery using spiking experiments of MCF7, SKBR3 and MDA MB-231 breast cancer cell lines and different isolation methods such as ficoll density gradient, erythrolysis, the CellSearch platform and the ISET system. We have reported that ISET’s recovery rate was higher compared to CellSearch and ficoll density gradient [31]. A spot from every patient (10^6^ cells/spot) was used for the identification of CTCs and evaluation of the expression of JUNB and CXCR4, after triple immunofluorescence staining followed by VyCAP or confocal microscopy analysis.

### 2.4. Cytospin Preparation

Ten mL of peripheral blood in EDTA tubes was obtained from SCLC patients, and their peripheral blood mononuclear cells (PBMCs) were isolated using Ficoll-Hypaque (d = 1.077 g/mol) density centrifugation at 1800 rpm for 30 min without brakes. PBMCs were washed twice with PBS and centrifuged at 1500 rpm for 10 min. Aliquots of 500,000 cells/500 μL were centrifuged at 2000 rpm for 2 min on Superfrost glass slides (Thermo Fisher Scientific, Waltham, MA, USA). Cytospins were dried up and stored at −80 °C. Two slides from each patient were analyzed for the identification of CTCs and evaluation of the expression of JUNB and CXCR4, after triple immunofluorescence staining followed by confocal microscopy and VyCAP scanning microscopy.

### 2.5. Spiking Experiments

H1299 cells were spiked into healthy volunteers’ PBMCs (1000 H1299 cells/100,000 PBMCs) and centrifuged at 2000 rpm for 2 min on Superfrost glass slides (Thermo Fisher Scientific, Waltham, MA, USA). Slides from spiking experiments were used as controls in order to evaluate the sensitivity and specificity of the method/antibodies and simulate the CTC microenvironment in blood circulation.

### 2.6. Triple Immunofluorescence Staining

CTCs from ISET membranes of NSCLC patients, cytospin preparations of SCLC patients, and spiked samples, were analyzed for expression of cytokeratin (CK), JUNB and CXCR4 using triple immunostaining with the corresponding antibodies. Firstly, control and SCLC patients’ cytospins were incubated with PBS for 5 min followed by fixation with 3% paraformaldehyde for 30 min at RT and then permeabilization with 0.5% Triton X-100 for 10 min at RT. For NSCLC patients’ ISET spots, cells were hydrated with PBS for 5 min and permeabilized with 0.5% Triton X-100 for 10 min. Non-specific binding in cytospins was avoided by blocking with 5% FBS in PBS at 4 °C overnight, while on ISET spots blocking was achieved with 10% FBS in PBS for 1 h. After blocking, cells were incubated for 1 h with the A45-B/B3 mouse antibody (Ab) for the detection of the CKs 8/18/19 (Amgen, Southern Oaks, CA, USA) in SCLC cytospins and with a mixture of the A45-B/B3 mouse Ab with the CK7 mouse Ab (Invitrogen, Waltham, MA, USA), in order to increase the recovery rate of CTCs, as CK7 is highly expressed in NSCLC cells [1,9,10]. Alexa 555 anti-mouse was used as a secondary antibody (Life Technologies, Carlsbad, CA, USA) for 45 min. Samples were further incubated with the CXCR4 rabbit Ab (ABCAM, Cambridge, MA, USA) for 1 h and as a next step, they were stained with the Alexa 647 anti-rabbit Ab (Life Technologies, Carlsbad, CA, USA) for 45 min. Cells were then incubated for 1 h with the anti-JUNB mouse Ab conjugated with Alexa 488 (Santa Cruz Biotechnology, Santa Cruz, CA, USA). Finally, samples were mounted on Prolong antifade medium containing DAPI for nuclear visualization.

H1299 cells spiked in healthy volunteers’ PBMCs were used as positive and negative controls. Specifically, negative controls were prepared by omitting one of the first antibodies, but incubating the cells with the respective secondary antibody. Each experiment included three different negative controls and one positive for all the antibodies.

Slides were analyzed with the VyCAP system (VyCAP B.V., Enschede, the Netherlands) and a Leica TCS SP8 confocal microscope (Leica Microsystems, Wetzlar, Germany). Furthermore, slides were also indicatively double-checked using the ACCEPT software (automatic software for CTCs detection, University of Twente, Enschede, the Netherlands). Due to limited expression of CKs in lung cancer CTCs, especially in NSCLC [1,10,11], apart from CK-staining, cytomorphological criteria described by Meng et al. [32] (such as high nuclear/cytoplasmic ratio and larger cells than white blood cells) were applied in order to characterize a cell as a CTC. Finally, the identification of CTCs was performed blindly to clinical data.

### 2.7. Statistical Analysis of the Clinical Data

Statistical tests were performed at the 5% level of significance and the statistical software that was used for the analysis was SPSS version 27 (IBM, Armonk, NY, USA). Overall survival (OS) was defined as the time period from enrollment to the study, until death from any cause or the last time follow-up that the patient was reported alive. Progression-free survival (PFS) was defined as the time-space between the enrolment to the study, and disease relapse or death, whatever occurred first. Kaplan–Meier analysis was used for both NSCLC and SCLC samples to correlate the presence of CTCs, or specific phenotypes such as JUNB-/CXCR4-positive with patients’ clinical outcome. Cox regression was also performed for the analysis of the CTCs’ number per phenotype regarding patients’ OS and PFS. Kaplan–Meier curves and Cox regression analysis for PFS and OS were compared using the log-rank test. Statistical tests between the mean percentages of phenotypes for each lung cancer subtype were carried out with Wilcoxon signed-rank nonparametric tests. Multivariate analysis for NSCLC samples, using Cox regression analysis, was also used; however, no statistically significant results could be obtained. SCLC patients were subgrouped into those with extensive disease, with cut-offs of ≥1, ≥2, ≥3, and ≥4 CTCs for every phenotype. Mann–Whitney tests were performed between NSCLC and SCLC samples for the observed phenotypes. Spearman’s analysis was also used to correlate the number of CTCs per phenotype. Finally, all clinicopathological data for patients enrolled in this study were analyzed to identify a possible association with patients’ outcome; however, none of these were statistically significant.

## 3. Results

### 3.1. JUNB and CXCR4 Expression in CTCs Derived from NSCLC Patients

The expression of JUNB and CXCR4 in NSCLC CTCs was evaluated by immunofluorescence using antibodies for CK, JUNB and CXCR4 (Figure 1 and Appendix A).

Thirty NSCLC patients were analyzed before the initiation of chemotherapy (at baseline). CTCs were detected in 53% patients (16 out of 30). Among the CK-positive NSCLC patients, 50% (8 out of 16) had the (CK+/JUNB+/CXCR4+) phenotype, 44% (7 out of 16) had the (CK+/JUNB+/CXCR4–) phenotype, 6% (1 out of 16) had the (CK+/JUNB–/CXCR4+) phenotype and 38% (6 out of 16) had the (CK+/JUNB–/CXCR4–) phenotype (Figure 2a). Some patients had more than one phenotype; 25% (4 out of 16 patients) had two of the four identified phenotypes, while 6% (1 out of the 16 patients) had three. However, most of the CK-positive patients (69%, 11 out of 16) only had one detectable phenotype (Appendix A).

In terms of the total identified CTCs, 42% were characterized as (CK+/JUNB+/CXCR4+), 33% as (CK+/JUNB+/CXCR4–), 6% as (CK+/JUNB–/CXCR4+), and 18% were classified as (CK+/JUNB–/CXCR4–). The percentage of (CK+/JUNB–/CXCR4+) CTCs was significantly lower compared to (CK+/JUNB+/CXCR4+) CTCs (Wilcoxon test: *p* = 0.018) and (CK+/JUNB+/CXCR4–) CTCs (Wilcoxon test: *p* = 0.024) (Figure 2b).

Regarding JUNB and CXCR4 alone, 88% the patients with identified CTCs (14 out of 16) were CK+/JUNB+ and 56% (9 out of 16) were CK+/CXCR4+ (Figure 2c). Of the total observed CTCs in NSCLC patients, 75% were JUNB-positive, while 48% were CXCR4-positive. The percentage of JUNB-positive CTCs was significantly higher compared to CXCR4-positive CTCs (Wilcoxon test: *p* = 0.024) (Figure 2d).

### 3.2. Clinical Relevance in NSCLC Patients

Survival analysis revealed that the existence of at least one CTC in NSCLC patients’ samples was associated with lower PFS (Kaplan–Meier, Log Rank, *p* = 0.015, HR = 4.46), compared to NSCLC patients without detectable CTCs (3.6 months with range 1.3–5.8 vs. 9.3 months with range 5.5–13.2) (Figure 3a). Furthermore, the detection of at least ≥2 CTCs in NSCLC patients was associated with worse OS (Kaplan–Meier, Log Rank, *p* = 0.02, HR = 4.94) compared to the rest of NSCLC patients without detectable CTCs (4 months with range 1.6–6.3 vs. 9.4 months with range 6.8–12) (Figure 3b).

The detection of (CK+/JUNB+/CXCR4+) CTCs in NSCLC patients was associated with poorer PFS (Kaplan–Meier, Log Rank, *p* = 0.007, HR = 5.21; Figure 3c) compared to patients without this phenotype (0.8 months with range 0–1.8 vs. 7.7 months with range 4.9–10.4). Additionally, the detection of at least ≥2 (CK+/JUNB+/CXCR4+)-CTCs was correlated to decreased OS (Kaplan–Meier, Log Rank, *p* < 0.001, HR = 2.16) compared to NSCLC patients with ≤1 (0.7 months with range 0–1.3 vs. 9 months with range 6.7–11.3) (Figure 3d). Cox regression analysis confirmed these results (PFS: *p* = 0.017, HR = 1.66 and OS: *p* = 0.011, HR = 2.16).

The existence of JUNB-positive CTCs was also associated with decreased PFS (Kaplan–Meier, Log Rank, *p* = 0.041, HR = 3.22) compared to patients without detectable CTCs (3.8 months with range 1.4–6.3 vs. 8.4 months with range 4.5–12.2) (Figure 3e). Moreover, the existence of at least ≥2 JUNB-positive CTCs in patients’ samples was associated with worse OS (Kaplan–Meier, Log Rank, *p* = 0.002, HR = 6.86, 3 months with range 0.4–5.6 vs. 9.5 months with range 7–12) (Figure 3f). Cox regression analysis for the number of JUNB-positive CTCs revealed poorer PFS (*p* = 0.028, HR = 1.62) and OS (*p* = 0.01, HR = 1.91).

Finally, the detection of CXCR4-positive CTCs was associated with lower PFS (Kaplan–Meier, Log Rank, *p* = 0.007, HR = 5.21) compared to patients with no CXCR4-positive CTCs (0.8 months with range 0–1.8 vs. 7.7 months with range 4.9–10.4) (Figure 3g). In addition, the identification of at least ≥2 CXCR4-positive CTCs in patients’ samples was associated with decreased OS (Kaplan–Meier, Log Rank, *p* < 0.001, HR = 2.16, 0.7 months with range 0–1.3 vs. 9 months with range 6.7–11.3) (Figure 3h). Cox regression analysis further confirmed this observation (PFS: *p* = 0.017, HR = 1.66 and OS: *p* = 0.011, HR = 2.16).

### 3.3. JUNB and CXCR4 Expression in CTCs Derived from SCLC Patients

The expression of JUNB and CXCR4 in SCLC CTCs was evaluated by immunofluorescence using antibodies for CK, JUNB and CXCR4 (Figure 4).

Thirty-seven SCLC patients at baseline were analyzed and CTCs were observed in 84% of them (31 out of 37). Among the patients with detectable CTCs, 71% (22 out of 31) patients had the (CK+/JUNB+/CXCR4+) phenotype, and the same percentage of patients had the (CK+/JUNB+/CXCR4–) and the (CK+/JUNB–/CXCR4+) phenotypes, while 84% (26 out of 31) had the (CK+/JUNB–/CXCR4–) phenotype (Figure 5a). Furthermore, SCLC patients presented high phenotypic heterogeneity as most of them either had all four observed phenotypes (39%, 12 out of 31 patients) or three out of the four (29%, 9 out of 31 patients) phenotypes. Regarding the rest of the patients, 23% (7 out of 31) either had two or only had one phenotype (10%, 3 out of 31 patients) (Appendix A).

Considering the total detected CTCs, 19% were identified as (CK+/JUNB+/CXCR4+), 27% as (CK+/JUNB+/CXCR4–), 14% as (CK+/JUNB–/CXCR4+), and the rest (41%) were classified as (CK+/JUNB–/CXCR4–). The percentage of (CK+/JUNB–/CXCR4–) CTCs was significantly higher in comparison to (CK+/JUNB+/CXCR4+) CTCs (Wilcoxon test: *p* = 0.008), to (CK+/JUNB+/CXCR4–) CTCs (Wilcoxon test: *p* = 0.025) and to (CK+/JUNB–/CXCR4+) CTCs (Wilcoxon test: *p* < 0.001) (Figure 5b).

Distinct analysis of each examined molecule revealed that from the patients with detectable CTCs, 94% (29 out of 31) harbored (CK+/JUNB+) CTCs and 84% (26 out of 31) harbored (CK+/CXCR4+) CTCs (Figure 5c). Of the total identified CTCs in SCLC patients, 45% were JUNB-positive, while 32% were CXCR4-positive (Figure 5d).

It is interesting to notice from the three CK-positive SCLC patients, who only had one phenotype detected in their CTCs, two of them had (CK+/JUNB+/CXCR4–) and one had (CK+/JUNB+/CXCR4+) CTCs. Moreover, despite the fact that (CK+/JUNB–/CXCR4–) phenotype was the most frequent among patients, none of them had exclusively (CK+/JUNB–/CXCR4–) CTCs. In addition, none of the SCLC patients had exclusively (CK+/JUNB–/CXCR4+)-CTCs (Appendix A).

### 3.4. Clinical Relevance in SCLC Patients

The presence of ≥4 CXCR4-positive CTCs was a poor prognostic factor for OS in SCLC patients with extensive disease (Kaplan Meier, Log Rank, *p* = 0.041, HR = 5.01; Figure 6a), compared to SCLC patients with ≤3 CXCR4-positive CTCs (4.5 months with range 1.6–7.4 vs. 11.4 months with range 8.5–14.4).

No statistical clinical relevance has been identified regarding the rest of the examined molecules and the corresponding phenotypes in SCLC patients.

## 4. Discussion

Low expression of epithelial markers in CTCs [1,7,9,10,11] due to EMT has made their detection challenging and this holds true for CTCs isolated from lung cancer patients. In this study, CTCs were detected in the majority of SCLC patients (84%), while in NSCLC only half of the patients (53%) were CK-positive. The positivity rates reported are in line with previous studies in NSCLC [10] and SCLC [33], and a bit higher compared to certain other studies in SCLC [34]. In the current study the presence of at least one CTC was associated with lower PFS (Kaplan–Meier, Log Rank, *p* = 0.015, HR = 4.46) in NSCLC patients. This is in agreement with previous studies in NSCLC [35,36]. However, in SCLC no clinical relevance has been found regarding the total number of CTCs. These findings indicate that enumeration is not always informative, regarding prognostic value.

Conversely, the study of the heterogeneity [5,16,17,33] and metastatic potential [6] of lung cancer could possibly be more enlightening regarding prognosis. Therefore, increasing interest is now towards the identification of new biomarkers and the elucidation of their role in patients’ clinical outcome for the subsequent categorization of patients for the appropriate therapeutic regime. Several studies have investigated the expression profiles and role of the transcription factor JUNB in tumorigenesis in NSCLC, exploring the possible mediating pathways [23,27,37]. *JUNB* expression was increased in NSCLC tissues compared to healthy lung tissues [23]. Regulation of *JUNB* expression was proposed to be mediated by the XCR1/XCL1 axis, introducing a different chemokine-related pathway which could be implicated in NSCLC progression [37]. CXCR4 is a G protein-coupled receptor, member of the chemokine receptors. CXCR4 signaling influences a variety of pathways. Through these pathways, CXCR4 exerts its pleiotropic roles in both physiological and pathological conditions [20,38]. CXCR4 was highly expressed in NSCLC primary tumors and metastatic lesions on tissue microarrays [28], while a recent study reveals a possible pathway of CXCR4/STAT3/Slug implicated in radio resistance on NSCLC cells [39]. Both JUNB and CXCR4 have been related to metastasis [20,21,22,23,24,26,37,40], while to date, evaluation of both JUNB and CXCR4 in lung cancer patients’ CTCs has not been presented.

In the current study, JUNB and CXCR4 were both expressed in CTCs of NSCLC, with JUNB expression being higher than CXCR4. More precisely, 88% of NSCLC patients were JUNB-positive, while 56% of NSCLC patients were CXCR4-positive. This high expression level of both JUNB and CXCR4 was also observed in our previous studies for metastatic breast cancer patients’ CTCs (JUNB: 65% and CXCR4: 90% in patients’ cohort) and for DTCs from the bone marrow of early breast cancer patients (JUNB: 95% and CXCR4: 92% in patients’ cohort) [18,19]. However, in the present study, the percentage of patients having CTCs with JUNB expression was higher than those with CXCR4 compared to breast cancer cases, implying that JUNB could be a very important player in metastatic procedure in NSCLC patients. Regarding district phenotypes, the predominant phenotype in NSCLC was the (CK+/JUNB+/CXCR4+) in 50% of patients, followed by the (CK+/JUNB+/CXCR4–) in 44%, the (CK+/JUNB–/CXCR4–) in 38% and the (CK+/JUNB–/CXCR4+) in 6% of patients. As was seen previously [19], the (CK+/JUNB+/CXCR4+) phenotype was also the most predominant phenotype (90%) in DTCs from breast cancer patients, followed by the (CK+/JUNB+/CXCR4–) in 36% and the (CK+/JUNB–/CXCR4–) in 31%, while the (CK+/JUNB–/CXCR4+) was the less frequent (5%), concluding that the phenotypic patterns were similar in both NSCLC and breast cancer cases. Furthermore, in this study, the majority (69%) of the CK-positive NSCLC patients, only had one out of the four detected phenotypes and only one patient (6%) had three phenotypes. However, in our previous study, 54% of breast cancer patients had more than one phenotype in their DTCs [19] implying different expression levels of the examined biomarkers in transition from CTCs to DTCs. Therefore, this dynamic expression profile could enhance the role of JUNB and CXCR4 in invasion, establishment of cancer cells in a new tissue and metastasis, confirming related studies [20,21,22,23,24,26,37,40].

This observation is reinforced by the clinical significance of (CK+/JUNB+/CXCR4+), JUNB+ and CXCR4+ phenotypes. Particularly, the presence of the (CK+/JUNB+/CXCR4+) phenotype in NSCLC patients was linked to poorer PFS (Kaplan–Meier, Log Rank, *p* = 0.007, HR = 5.21, Cox regression: *p* = 0.017, HR = 1.66) and OS (Cox regression: *p* = 0.011, HR = 2.16). Furthermore, the existence of ≥2 (CK+/JUNB+/CXCR4+) CTCs in NSCLC patients was associated with worse OS (Kaplan–Meier, Log Rank, *p* < 0.001, HR = 2.16). The current study also found that the presence of JUNB-positive CTCs and CXCR4-positive CTCs in NSCLC patients were correlated with shorter PFS (Kaplan–Meier, Log Rank, *p* = 0.041, HR = 3.22, Cox regression: *p* = 0.028, HR = 1.62, and Kaplan–Meier, Log Rank, *p* = 0.007, HR = 5.21, Cox regression: *p* = 0.017, HR = 1.66, respectively) and OS (Cox regression: *p* = 0.01, HR = 1.91, and Cox regression: *p* = 0.011, HR = 2.16 respectively), reflecting the importance of both biomarkers in clinical practice. Moreover, the detection of ≥2 JUNB-positive CTCs and CXCR4-positive CTCs in NSCLC patients were correlated with lower OS (Kaplan–Meier, Log Rank, *p* = 0.002, HR = 6.86, and Kaplan–Meier, Log Rank, *p* < 0.001, HR = 2.16, respectively). Interestingly, in line with these data the presence of JUNB in breast cancer patients’ CTCs has been previously associated with significantly lower median PFS (Kaplan–Meier analysis, *p* = 0.015) and OS (Cox regression, *p* = 0.026, HR = 2.31, and Kaplan–Meier analysis, *p* = 0.02) [18]. In multivariate analysis in the same study, JUNB was also identified as an independent prognostic factor of OS (*p* = 0.016, HR = 2.25) [18]. The same observation was also made in early breast cancer patients’ DTCs. Particularly, DTCs, with the phenotype (CK+/JUNB+/CXCR4+), were correlated with lower OS (Cox regression, *p* = 0.023, HR = 1.03) [19]. These observations are also in agreement with other studies in NSCLC tissues, where high JUNB expression using sequencing data from the Cancer Genome Atlas was correlated with decreased OS of 3 years [23]. In terms of CXCR4, flow cytometric analysis of NSCLC patients’ CTCs, revealed that low expression of CXCR4 and CK was associated with better OS [28]. Several meta-analyses of CXCR4 expression in NSCLC [41,42,43,44] have suggested that increased CXCR4 expression is correlated with shorter OS, shorter disease-free survival, advanced TNM stages, and metastasis.

In the present study, JUNB and CXCR4 were also both expressed in CTCs of SCLC patients, with the expression of JUNB being higher than CXCR4. More precisely, 94% of SCLC patients were JUNB-positive, while 84% of SCLC patients were CXCR4-positive. These high percentages of both molecules suggest an important role of CXCR4 and JUNB in the metastatic pathway of SCLC patients. The predominant phenotype in SCLC was the (CK+/JUNB–/CXCR4–) in 84% of patients, followed by the other three identified phenotypes, which were equally distributed in 71% of patients. However, it is noteworthy, that although the (CK+/JUNB–/CXCR4–) phenotype was the most frequent in SCLC, none of the patients nor their CTCs had exclusively that phenotype, which was also the case in breast cancer DTCs [19]. Moreover, in the CK-positive SCLC patients, the minority (10%) only had one phenotype, while a large proportion (68%) had three or four phenotypes. This observation highlights the heterogeneity of CTCs in the blood of SCLC patients, indicating the necessity of identifying the most aggressive subclone among these cells. To this end, analysis of the clinical significance of these subclones revealed that the presence of ≥4 CTCs positive for CXCR4 was related to poorer OS (Kaplan–Meier, Log Rank, *p* = 0.041, HR = 5.01) in extensive disease patients. These results, resemble an earlier study whereby ≥7% of CXCR4-positive CTCs in SCLC patients with extensive-stage disease at baseline was associated with shorter PFS, but not OS [30]. In addition, SCLC tumors expressing CXCR4, and the urokinase-type plasminogen activator receptor (uPAR) were found to have worse prognosis [45]. Furthermore, in SCLC, a meta-analysis has suggested a correlation between CXCR4 expression and unfavorable OS, albeit not statistically significant [44].

Comparing findings between NSCLC and SCLC suggests that (i) CTCs express both JUNB and CXCR4 in the two basic lung cancer subtypes, with JUNB expression being higher than CXCR4, (ii) SCLC patients’ rates for the four observed phenotypes are higher compared to NSCLC, (iii) exclusively for the JUNB-positive CTCs, the percentage was lower in SCLC, (iv) the predominance and abundance of the phenotypic patterns of (CK/JUNB/CXCR4) are different among patients and their total CTCs, with the NSCLC cases being more similar to breast cancer cases of our previous studies [18,19] and (v) the heterogeneity in terms of phenotypes was more pronounced in SCLC (the vast majority of patients had 3 or 4 phenotypes). All the above highlight the differences between the two lung types and are suggestive of the increased complexity of SCLC. This is in line with other studies in SCLC, where different expression profiles, in terms of the examined biomarkers, have been reported [5,33,34,46,47,48].

Overall, JUNB and CXCR4 are both highly expressed in CTCs from NSCLC and SCLC patients and correlated to patients’ clinical outcome. However, we should clarify that this is a pilot study, and the present results must be confirmed in a larger patient cohort. Further, studies should include more patients even at different time points of treatment, providing information about the prognostic and predictive relevance of these proteins. Although the number of patients in this manuscript is relatively small, it provides the basis for the importance of JUNB and CXCR4 expression in lung cancer patients’ CTCs as an indication of patients’ poor survival and further supports their role in metastasis, confirming related studies [20,21,22,23,24,26,37,40]. It is also noteworthy that several current clinical trials (NCT04177810, NCT02907099) in pancreatic cancer include anti-PD-1/PD-L1 therapies in conjunction to anti-CXCR4 therapies. Analysis of PD-L1 and CXCR4 could hence be informative in the future in lung cancer patients.

## 5. Conclusions

Tumor heterogeneity is implicated in treatment resistance and cancer relapse. JUNB and CXCR4 are expressed in lung cancer patients and their expression correlates with worse prognosis. The present study portrays JUNB and CXCR4 as potentially interesting biomarkers for the identification of NSCLC and SCLC patients at higher risk.

## Figures and Tables

**Figure 1 cancers-15-00171-f001:**
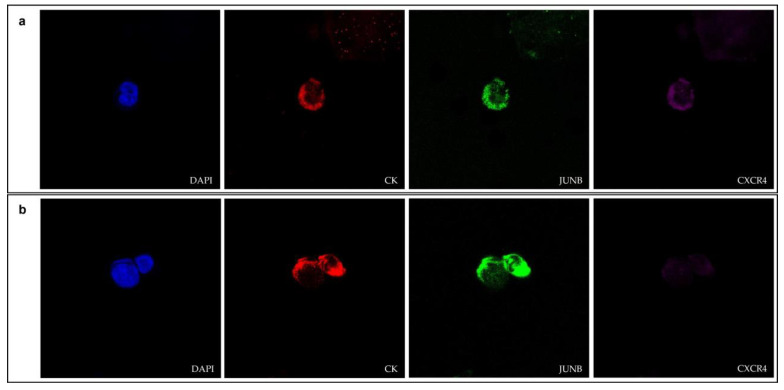
(**a**,**b**) Cytokeratin (red), JUNB (green) and CXCR4 (purple) expression in CTCs isolated from NSCLC patients (confocal microscopy). Nuclei (blue) were stained with DAPI. Magnification 40×.

**Figure 2 cancers-15-00171-f002:**
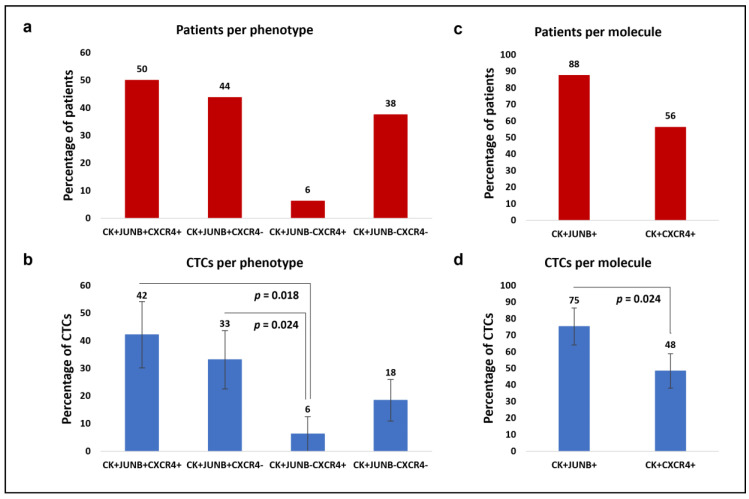
Phenotypic patterns of (CK/JUNB/CXCR4) expression in NSCLC blood samples. (**a**) Percentages of patients with (CK+/JUNB+/CXCR4+), (CK+/JUNB+/CXCR4–), (CK+/JUNB–/CXCR4+) and (CK+/JUNB–/CXCR4-) phenotypes. (**b**) Average percentages of the total CTCs with the identified (CK+/JUNB+/CXCR4+), (CK+/JUNB+/CXCR4–), (CK+/JUNB–/CXCR4+) and (CK+/JUNB–/CXCR4–) phenotypes. (**c**) Percentages of patients with (CK+/JUNB+) and (CK+/CXCR4+) phenotypes. (**d**) Average percentages of the total CTCs with (CK+/JUNB+) and (CK+/CXCR4+) phenotypes.

**Figure 3 cancers-15-00171-f003:**
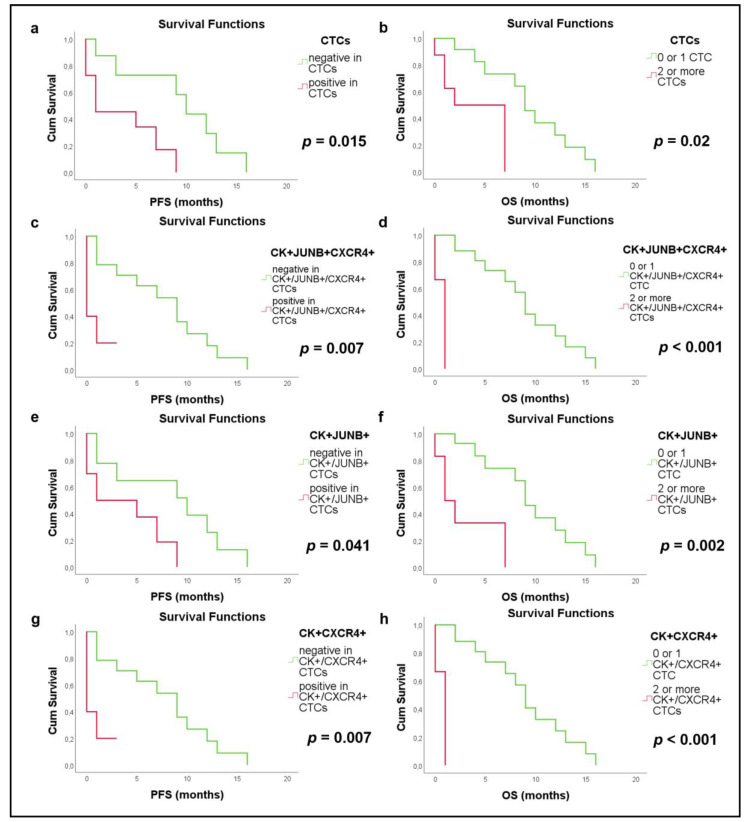
Kaplan–Meier survival curves in NSCLC patients related to CTCs and specific phenotypes. (**a**) Progression free survival (PFS) for detection of at least one CTC (*p* = 0.015, HR = 4.46) (**b**) Overall survival (OS) for detection of at least two CTC (*p* = 0.02, HR = 4.94) (**c**) PFS for detection of the (CK+/JUNB+/CXCR4+) phenotype (*p* = 0.007, HR = 5.21) (**d**) OS for detection at least two (CK+/JUNB+/CXCR4+) CTCs (*p* < 0.001, HR = 2.16) (**e**) PFS for identification of (CK+/JUNB+) CTCs (*p* = 0.041, HR = 3.22) (**f**) OS for identification of (CK+/JUNB+) CTCs (*p* = 0.002, HR = 6.86) (**g**) PFS for identification of (CK+/CXCR4+) CTCs (*p* = 0.007, HR = 5.21) (**h**) OS for identification of (CK+/ CXCR4+) CTCs *p* < 0.001, HR = 2.16).

**Figure 4 cancers-15-00171-f004:**

Cytokeratin (red), JUNB (green) and CXCR4 (purple) expression in a CTC isolated from a SCLC patient (VyCAP imaging system). Nuclei (blue) were stained with DAPI. Magnification 40×.

**Figure 5 cancers-15-00171-f005:**
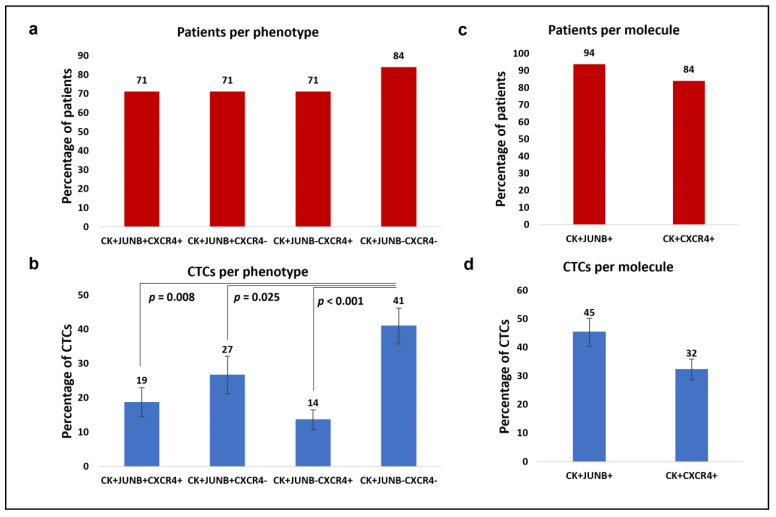
Phenotypic patterns of (CK/JUNB/CXCR4) expression in SCLC blood samples. (**a**) Percentages of patients with (CK+/JUNB+/CXCR4+), (CK+/JUNB+/CXCR4–), (CK+/JUNB–/CXCR4+) and (CK+/JUNB–/CXCR4-) phenotypes. (**b**) Average percentages of the total CTCs with the identified (CK+/JUNB+/CXCR4+), (CK+/JUNB+/CXCR4–), (CK+/JUNB–/CXCR4+) and (CK+/JUNB–/CXCR4–) phenotypes. (**c**) Percentages of patients with (CK+/JUNB+) and (CK+/CXCR4+) phenotypes. (**d**) Average percentages of the total CTCs with (CK+/JUNB+) and (CK+/CXCR4+) phenotypes.

**Figure 6 cancers-15-00171-f006:**
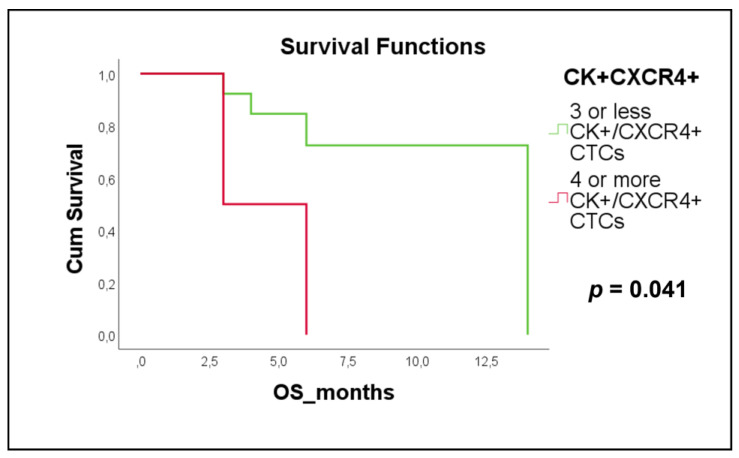
Kaplan–Meier survival curves for OS in SCLC patients with extensive disease related to the detection of CK+/CXCR4+ CTCs with cut-off at least 4 (*p* = 0.041, HR = 5.01).

**Table 1 cancers-15-00171-t001:** NSCLC patients’ clinical characteristics.

Characteristics	Sub-Categories	Values
Median age		66 years (range 44–82 years)
Stage	III	6 (20%)
	IV	24 (80%)
Histology	Adenocarcinoma	18 (60%)
	Squamous cell carcinoma	12 (40%)
EGFR	WT	11 (37%)
	ND	19 (63%)

NSCLC, non-small-cell lung cancer; WT, wild type; ND, not defined.

**Table 2 cancers-15-00171-t002:** SCLC patients’ clinical characteristics.

Characteristics	Sub-Categories	Values
Median age		68 years (range 44–79 years)
Stage	Limited	5 (14%)
	Extensive	20 (54%)
	Unknown	12 (32%)
Smoking	Yes	29 (78%)
	No	0 (0%)
	Unknown	8 (22%)
Family history of cancer	Yes	8 (22%)
	No	21 (57%)
	Unknown	8 (22%)
Metastasis	CNS	10 (27%)
	Liver	8 (22%)
	Pancreas	1 (3%)
	Bones	5 (14%)
	Adrenal glands	3 (8%)
	LNs	5 (14%)
	Unknown	14 (38%)

SCLC, small-cell lung cancer; CNS, central nervous system; LNs, lymph nodes.

## Data Availability

Data presented in the study are available upon request from the corresponding author.

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
