# Peer review of "Phenotypic Characterization of Circulating Tumor Cells Isolated from Non-Small and Small Cell Lung Cancer Patients"

_cancers, 2022, doi:10.3390/cancers15010171_

Round 1
Reviewer 1 Report
This research article titled “Phenotypic Characterization of Circulating Tumor Cells Isolated from Non-Small and Small Cell Lung Cancer Patients” authored by Roumeliotou and colleagues is very interesting and informative. They have isolated CTSs from patients using a platform called ISET and then analyzed by confocal microscopy.
1. What was the time laps between blood draw and CTC isolation? What kind of blood collection tubes were used for this study? Since CTCs are very fragile this information is important.
2. Please provide ISET platform manufacturer’s information in this section so that anybody who want to reproduce your results will have that information.
3. Do you have any published information about ISET platform’s efficiency related to CTC recovery from blood? Have you checked this using spiked cancer cells? This information is useful for research who doing similar kind of work.
4. Figure 2
Y-axis in figure 2, please remove coma and two decimal points, It looks like 10, 00, 20,00. Or use period instead of coma. Current format is very confusing.
5. Figure 4.
Cytokeratin is a marker for tumor cells. Why in this figure only one cell is positive for Cytokeratin while there are many cells that are positive for JUNB and CXCR4?
6. Figure 5.
Please fix the Y-axis in this figure by removing coma and two decimal points.
Author Response
REVIEWERS’ COMMENTS
We thank the reviewer for the comments. Please find below our point-by-point answers.
REVIEWER 1:
- What was the time laps between blood draw and CTC isolation? What kind of blood collection tubes were used for this study? Since CTCs are very fragile this information is important.
Indeed, CTCs are very fragile and tend to rapidly degrade. We have therefore initially used CellSave (Menarini) tubes for optimized cell preservation. However, we observed worse viability compared to cells from EDTA K2 tubes, hence leading us to choose the latter for collecting our blood samples. Optimized cell viability is of paramount importance for the lab, since isolated CTCs are further cultured for functional analysis in parallel lab projects. The following sentence is added in section 2.2, line126: “Peripheral blood (10 ml), from all patients as well as from 4 healthy donors, was collected in EDTA K2 tubes. In section 2.3 (lines 140-142), we have also added "Isolation of CTCs and all experimental procedures were performed immediately after collection or at most within 24 h of blood collection."
- Please provide ISET platform manufacturer’s information in this section so that anybody who want to reproduce your results will have that information.
Section 2.3 has been enhanced with additional information regarding the ISET technology (lines 140-160). “Isolation of CTCs from NSCLC is challenging because of the extensive loss of epithelial markers on their surface and their fewer numbers in the blood. Due to these difficulties, isolation of CTCs from NSCLC blood samples was performed using the ISET (Isolation based on the Size of Tumor Cells) technology. The method, which is very well established in our lab according to our previous publications (Kallergi et al., 2016; Katsarou et al., 2022), is label-independent and can successfully capture CTCs which present heterogeneous phenotypic profiles. Isolation/filtration relies on the larger size of CTCs compared to the majority of leukocytes, increasing the recovery rate of CTCs [Kallergi et al. 2016]. CTCs were isolated according to the manufacturer’s instructions. For this purpose, 10 ml of peripheral blood in EDTA tubes was diluted in 1:10 ISET buffer (Rarecells, Paris, France) for 10 min at room temperature (RT). This application-specific buffer performs erythrolysis and contains formaldehyde to preserve the integrity of CTCs. The diluted samples were filtered through the ISET membrane using depression adjusted at 10 kPa, whereby fixed cells that cannot pass through the pores (i.e. CTCs) are retained on the membrane, forming 10 spots. The method’s detection threshold based on the manufacturing company is 1 CTC per 10 ml of blood”.
- Do you have any published information about ISET platform’s efficiency related to CTC recovery from blood? Have you checked this using spiked cancer cells? This information is useful for research who doing similar kind of work.
The following text was added in section 2.3, lines 156-160. “We have previously evaluated CTCs’ recovery using spiking experiments of MCF7, SKBR3 and MDA MB-231 breast cancer cell lines and different isolation methods such as ficoll density gradient, erythrolysis, the CellSearch platform, and the ISET system. We have reported that ISET’s recovery rate was higher compared to CellSearch and ficoll density gradient (Kallergi et al., 2016)”.
- Figure 2
Y-axis in figure 2, please remove coma and two decimal points, It looks like 10, 00, 20,00. Or use period instead of coma. Current format is very confusing.
The format of the numbering of Y-axis has been changed in Fig.2.
- Figure 4.
Cytokeratin is a marker for tumor cells. Why in this figure only one cell is positive for Cytokeratin while there are many cells that are positive for JUNB and CXCR4?
We thank the reviewer for this very valid point. It is true that JUNB expression in Figure 4 can be found (in lower level) in neighboring cells, however, we count as CTC only the cell that revealed cytokeratin expression. It is possible that some of the JUNB-positive cells could also be cancer cells, but to be objective and not to overestimate the number of cancer cells in the bloodstream, we did not count these CK-negative/JUNB-positive cells as CTCs. The non-CTC cells, which are stained positive for JUNB and/or CXCR4 are not of immediate interest.
Furthermore, in parallel experiments of SCLC patients’ samples stained for CK/PD-L1/CD45, confirmed the existence of CTCs in the pool of contaminating blood cells.
- Figure 5.
Please fix the Y-axis in this figure by removing coma and two decimal points.
Similarly, changes have been made in the format of the numbering of Y-axis in Fig.5, according to the reviewer’s comment.
- Figure 5.
Please fix the Y-axis in this figure by removing coma and two decimal points.
Similarly, changes have been made in the format of the numbering of Y-axis in Fig.5, according to the reviewer’s comment.
We remain at your disposal for any further information
Sincerely,
Galatea Kallergi

Reviewer 2 Report
In this study, authors evaluated the expression of JUNB and CXCR4 in circulating tumor cells (CTCs) of lung cancer patients and investigated their candidate biomarker roles for new treatments of NSCLC and SCLC. The authors declared that “JUNB and CXCR4 were expressed in CTCs from lung cancer patients, and associated with patients’ survival, underlying their key role in tumor progression”. That is an interesting and valuable work which is close to the clinical use and could be benefit for patients, both in the diagnosis and treatment. However, several flaws should be revised before publication.
1) Authors should supplement more words about how the resource is used in their study, and the detailed parameters in the process of data analysis for independent dataset.
2) The authors should handle with batch effects and statical power of different source of data in this study. E.g., only 30 and 37 patients were included in this study according to the statement of the first paragraph in the Methods section.
3) The resolution of figures may not meet the criterion of publish demand, especially for Figures 3-6.
4) Authors should take care of the academic figures, comma should not be represented in the P-value, e.g., Figure 2b
Author Response
We thank the reviewer for the comments. Please find below our point-by-point answers.
REVIEWER 2:
- Authors should supplement more words about how the resource is used in their study, and the detailed parameters in the process of data analysis for independent dataset.
We have now explained better in sections 2.2 and 2.3 the procedure of sample collection and preparation. In addition, we have analyzed all patients enrolled in this study for correlations between their survival and the presence of JUNB and CXCR4 in their CTCs. An overview of the analysis performed is now included in section 2.7 (lines 220-234). “Kaplan Meier analysis was used for both NSCLC and SCLC samples to correlate the presence of CTCs, or specific phenotypes such as JUNB-/CXCR4-positive with patients’ clinical outcome. Cox regression was also performed for the analysis of the CTCs’ number per phenotype regarding patients’ OS and PFS. Multivariate analysis for NSCLC samples, using Cox regression analysis, was also used, however, no statistically significant results could be obtained. Therefore, patients were subgrouped into those with extensive disease, with cut offs of ≥1, ≥2, ≥3 and ≥4 CTCs for every phenotype. Mann-Whitney tests were performed between NSCLC and SCLC samples for the observed phenotypes. Spearman’s analysis was also used to correlate the number of CTCs per phenotype. Finally, all clinicopathological data for patients enrolled in this study were analyzed to identify a possible association with patient outcome, however, none of these were statistically significant."
- The authors should handle with batch effects and statical power of different source of data in this study. E.g., only 30 and 37 patients were included in this study according to the statement of the first paragraph in the Methods section.
This is a valid comment. As it is clarified and further strengthened in the discussion (lines 473-480), “this is a pilot study, and the present results must be confirmed in a larger patient cohort. Further studies should include more patients even at different time points of treatment, providing information about the prognostic and predictive relevance of these proteins. Although the number of patients in this study is relatively small, it provides the basis for the importance of JUNB and CXCR4 expression in lung cancer patients’ CTCs as an indication of patients’ poor survival and further supports their role in metastasis, confirming related studies (Teixidó et. al. 2013, Mukherjee et. al. 2013, Gokulnath et. al. 2017, Pei et. al. 2018, Sundqvist et. al. 2018, Lian et. al. 2015, Wang et. al. 2015, Hyakusoku et. al. 2016)”, already mentioned in discussion.
- The resolution of figures may not meet the criterion of publish demand, especially for Figures 3-6.
Figures have been amended to meet publishing criteria. Tiff files will also be uploaded upon resubmission.
- Authors should take care of the academic figures, comma should not be represented in the P-value, e.g., Figure 2b
The corrections have been made in Fig2b and to the rest of the manuscript.
We remain at your disposal for any further information
Sincerely,
Galatea Kallergi

Round 2
Reviewer 2 Report
None